# Economic burden of pulmonary arterial hypertension in Switzerland

Yuki Tomonaga[1]*, Mona Lichtblau[2], Silvia Ulrich[2], Sabina A. Guler[3], Patrick Yerly[4], Benoît Lechartier[5], Jean-Marc Fellrath[6], Anne Bergeron[7], Louise Bondeelle[7], Silviu-Mihail Chirila[8], Andrea Favre-Bulle[9], Sandro Stoffel[10], Matthias Schwenkglenks[10,11]

1 Epidemiology, Biostatistics and Prevention Institute (EBPI), University of Zurich, Zurich, Switzerland, 2 Department of Pulmonary Medicine, University Hospital of Zurich (USZ), Zurich, Switzerland, 3 Department of Pulmonary Medicine, Allergology and Clinical Immunology, Inselspital, Bern University Hospital, University of Bern, Bern, Switzerland, 4 Department of Cardiology, Lausanne University Hospital (CHUV), Switzerland, 5 Department of Respiratory Medicine, Lausanne University Hospital (CHUV), Lausanne, Switzerland, 6 Service of Pulmonary Medicine, Réseau Hospitalier Neuchâtelois (RHNe), Neuchâtel, Switzerland, 7 Pneumology Department, University Hospital of Geneva (HUG), Geneva, Switzerland, 8 Silviu-Mihail Chirila, MD, Clinic for Pneumology, University Hospital Basel (USB), Basel, Switzerland, 9 Andrea Favre-Bulle, MSD, Lucerne, Switzerland, 10 Department of Public Health, Institute of Pharmaceutical Medicine (ECPM), University of Basel, Basel, Switzerland, 11 Health Economics Facility, Faculty of Medicine, University of Basel, Basel, Switzerland

* yuki.tomonaga@uzh.ch

## Abstract

### Objectives

Pulmonary arterial hypertension (PAH) is a rare, progressive condition associated with high morbidity and healthcare resource utilization. This study aimed to estimate the annual direct and indirect costs of PAH in Switzerland, from a societal perspective.

### Materials and methods

A cross-sectional cost-of-illness study was conducted across six Swiss PAH centres between April and December 2024. Adult patients with confirmed PAH (World Health Organization [WHO] Group 1) were invited to complete a standardized questionnaire on work productivity losses, informal care, and healthcare utilization outside the enrolling centre. Clinical data on hospitalizations, outpatient visits, diagnostics, and treatments at the enrolling centre were extracted from medical records. Disease-specific costs were calculated by multiplying resource use and work losses by Swiss-specific unit costs and extrapolated to one year. Estimates were stratified by WHO functional class (WHO-FC) and ESC/ERS 2022 risk strata.

### Results

Among 124 participants aged between 18 and 89 years, the mean disease-specific total annual cost per patient was €138,958. Direct healthcare costs represented

**Data availability statement:** The raw individual-level data underlying this study cannot be shared publicly due to ethical and legal restrictions. Pulmonary arterial hypertension is a rare disease in Switzerland, and the small sample size combined with detailed clinical information creates a substantial risk of patient re-identification. The Cantonal Ethics Committee approved the study under the condition that data would be used only for the purposes described in the original consent forms and would not be shared outside the research team. Swiss data protection regulations (FADP, HRA) further restrict the sharing of sensitive health data. De-identified aggregated data supporting the findings of this study may be made available from the corresponding author upon reasonable request and subject to approval by the Cantonal Ethics Committee (Kantonale Ethikkommission Zürich, info.kek@kek.zh.ch). This does not alter our adherence to PLOS ONE policies on sharing data and materials.

**Funding:** This study was funded by MSD, Luzern, Switzerland. Three authors (Yuki Tomonaga, Sandro Stoffel, and Matthias Schwenkglenks) were employed by academic institutions that received funding from MSD for the organization and management of the study, the analysis and interpretation of the results, and the preparation of the manuscript. One author (Andrea Favre Bulle) is employed by the funder of the study. Mona Lichtblau, Silvia Ulrich, Sabina Guler, Patrick Yerly, Benoît Lechartier, Jean Marc Fellrath, Anne Bergeron, Louise Bondeelle, and Silviu Mihail Chirila received a fee per included patient to compensate for the time required for data extraction. The funder reviewed the manuscript for issues related to patent applications or intellectual property rights. The funder had no role in the study design; the conduct of the study; the collection, management, analysis, and interpretation of the data; the preparation of the manuscript; or the decision to submit the manuscript for publication. The funder provided support in the form of salaries for authors but did not have any additional role in the study design, data collection and analysis, decision to publish, or preparation of the manuscript. The specific roles of these authors are articulated in the 'Author Contributions' section.

78.5% of this amount (€109,114), driven primarily by pharmacological treatment (65% of total costs). Indirect costs amounted to 21.5% (€29,844). Costs increased with disease severity, ranging from €81,957 in WHO-FC 1 to €166,569 in WHO-FC 4, and from €130,970 in ESC/ERS low risk to €291,728 in ESC/ERS high risk. The total national burden was estimated at €48.5 million annually.

## Conclusions

PAH imposes a substantial economic burden in Switzerland, largely due to treatment costs and productivity losses. These findings highlight the need for strategies to reduce disease progression and associated societal costs.

## Introduction

Pulmonary arterial hypertension (PAH) is a rare, progressive, and fatal disease that can occur at any age [1–4]. The pathophysiology of PAH involves pulmonary endothelial dysfunction, resulting in impaired production of endogenous vasodilators (e.g., nitric oxide and prostacyclin), overexpression of vasoconstrictors (e.g., endothelin-1), and the abnormal proliferation of pulmonary vascular smooth muscle cells in pulmonary arterioles, which results in progressive pulmonary vascular remodelling, increased pulmonary vascular resistance, and eventually right heart failure.

Advances in the understanding of PAH pathophysiology have led to the development of targeted therapies that modulate four principal pathways: the endothelin, nitric oxide, prostacyclin and activin-signalling pathways. Currently approved treatments include endothelin receptor antagonists (ERAs), phosphodiesterase type 5 inhibitors (PDE5i), soluble guanylate cyclase stimulators, prostacyclin analogues, or receptor agonists like activin / bone morphogenic protein receptor type 2 (BMPR2)-signalling inhibitors. These therapies are used either as monotherapy or in combination, depending on the patient's functional class and risk profile [2,5–7]. Despite these options, PAH remains an incurable condition associated with high morbidity, reduced quality of life, and a 5-year mortality rate of approximately 50% [8].

In Central Europe, according to recently published estimates from the Global Burden of Disease 1990–2021 project, the age-standardized prevalence of PAH is approximatively 22.8 cases per million population (95% uncertainty interval UI: 18.9–27.9), with an annual age-standardized incidence of 4.7 cases per million [9,10]. These values are lower if compared to those reported in a systematic literature review: according to national systematic registry data from centralised healthcare systems, the PAH prevalence was 47.6–54.7 per million, while the incidence was 5.8 per million [2]. Swiss-specific epidemiological data are limited, but PAH is similarly considered an orphan disease. National registry data, such as from the Swiss Pulmonary Hypertension Registry, suggest trends comparable to those seen in other Western European countries [11].

The economic burden of PAH is substantial. It includes direct healthcare costs—particularly those related to hospitalization and long-term pharmacotherapy—as

**Competing interests:** Yuki Tomonaga, Sandro Stoffel, and Matthias Schwenkglenks were employed by academic institutions that received financial support from MSD for organizing and conducting the study, analysing the collected data, and writing the report/manuscript. Andrea Favre Bulle is employed by the funder of the study (MSD). Mona Lichtblau, Silvia Ulrich, Sabina Guler, Patrick Yerly, Benoît Lechartier, Jean Marc Fellrath, Anne Bergeron, Louise Bondeelle, and Silviu Mihail Chirila received a fee per included patient to compensate for the time required for data extraction from the funder of the study (MSD). Mona Lichtblau reports grants, honoraria, or consulting fees from MSD, Johnson & Johnson, Gebro Pharma, and Orpha Swiss. Silvia Ulrich reports research grants, honoraria, or consulting fees from the Swiss National Science Foundation, Zurich and Swiss Lung League, EMDO Foundation, Orpha Swiss, Janssen SA, MSD SA, Gebro Pharma, Ideogen, and AstraZeneca (all unrelated to the present work). Sabina Guler reports grants, honoraria, or consulting fees from MSD, Johnson & Johnson, Gebro Pharma, and Orpha Swiss. Anne Bergeron reports grants, honoraria, or consulting fees from AstraZeneca, Sanofi, GSK, Novartis, OM Pharma, and Boehringer Ingelheim. Matthias Schwenkglenks reports grants, honoraria, or consulting fees from AbbVie, Bristol Myers Squibb, Novartis, Pfizer, Roche, and AstraZeneca. All other authors declare no conflicts of interest in relation to this manuscript. This does not alter our adherence to PLOS ONE policies on sharing data and materials.

well as indirect costs such as those due to loss of productivity and need for informal care. International data indicate that annual per-patient costs can exceed €50,000–100,000, largely driven by the high price of PAH-specific therapies and frequent healthcare utilization [12–15].

Considering differences in drug pricing, health insurance reimbursement rules, and healthcare utilization patterns across countries that may influence cost structures, understanding the national economic burden of PAH is relevant for health system planning, cost-effectiveness assessments, and policy decisions regarding the allocation of healthcare resources. This study aims to address the current knowledge gap by quantifying the direct and indirect costs associated with PAH in Switzerland, thereby informing both clinical practice and health policy.

## Materials and methods

### General approach

The cross-sectional study was conducted in two main phases. In the first phase, participating clinical centres mailed an invitation package to all their patients diagnosed with PAH. The package included a questionnaire, patient information sheet, informed consent form, and pre-addressed/stamped envelopes for document return. One clinical centre opted for on-site recruitment during routine follow-up visits. To minimize selection bias, all PAH patients attending these visits were consecutively invited to participate and received the study materials to complete at home. The questionnaire collected information on sources of indirect costs associated with PAH, particularly productivity losses, as well as healthcare utilization involving general practitioners or specialists outside the treating centres. To limit recall bias, the questions referred to the four weeks preceding questionnaire completion [16] Patients were instructed to return the signed informed consent form to their treating hospital (by mail or during a subsequent visit) and to send the completed questionnaire directly to the study coordination centre (EBPI, University of Zurich). In the second phase, data on healthcare use due to PAH were extracted from the medical records of all patients who returned the questionnaire. This extraction covered the six months prior to study inclusion.

Ethical approval was obtained from the responsible ethics committees on 23/04/2024 (BASEC-Nr. 2023-02352). Patient recruitment started on 26/04/2024 and ended on 10/12/2024.

### Study population

The study included adult patients (≥18 years) living in Switzerland with a confirmed diagnosis of PAH (World Health Organization [WHO] group 1) established at least six months prior to study inclusion, who were receiving care at one of six participating clinical centres. The time point of enrolment was defined by the date patients completed the study questionnaire.

Inclusion criteria were diagnosis of PAH confirmed by right-heart catheterization, treatment at a participating PAH centre, provision of informed consent, and ability to complete the questionnaire in German, French, Italian, or English. Included PAH

subtypes were: idiopathic, heritable, drug- and toxin-induced, and PAH associated with connective tissue disease, HIV, portal hypertension, congenital heart disease, schistosomiasis, long-term response to calcium channel blockers, and those with features of pulmonary veno-occlusive disease and/or pulmonary capillary haemangiomatosis. Patients were excluded if they had pulmonary hypertension falling into WHO groups 2–5.

## Patient survey

The questionnaire used was based on the Productivity Cost Questionnaire (iPCQ), developed by the Institute for Medical Technology Assessment (iMTA) at Erasmus University, Rotterdam [16]. The iPCQ is a standardized tool designed to measure health-related productivity losses and comprises 18 items. Nine of these items gather demographic data and information on respondents' employment status—such as weekly hours of paid work, working days per week, and productivity loss over the past four weeks.

For the purposes of this study, the questionnaire was adapted to the specific context of PAH and expanded to include additional items. The adaptations concerned presenteeism (i.e., reduced productivity while at work) due to PAH, time lost for household chores, and the provision of formal and informal care by relatives both below and above retirement age.

The final adapted instrument consisted of 17 questions and captured a range of indirect cost domains, including reduced employment levels, absenteeism, presenteeism, reliance on formal care, informal caregiving (by both working-age and retired individuals), and limitations in performing household chores within the preceding four weeks. Additionally, the questionnaire recorded healthcare resource use outside the participating clinical centres, specifically the number of ambulatory visits to general practitioners or specialists within the preceding four weeks. The estimated time to complete the questionnaire was 10 minutes. The iMTA was informed about the questionnaire adaptations. Although the changes did not affect the questionnaire integrity, iMTA can't guarantee the accuracy or validity of the additionally collected data.

## Medical records data

Six clinical centres located in German-speaking and French-speaking Switzerland participated in the study. Clinical data extracted from medical records included patient characteristics (age, sex, WHO functional class [WHO-FC], risk stratification according to the guidelines developed in 2022 by the European Society of Cardiology (ESC) and the European Respiratory Society (2022 ESC/ERS Guidelines), and major co-morbidities) as well as healthcare resource utilization (hospitalizations related to PAH, length of stay, outpatient visits, emergency department visits, diagnostic procedures, and PAH-specific treatments) [7].

All data were recorded using standardized Microsoft Excel templates. Pilot data extractions in two participating centres confirmed the templates to be suitable. Medical record data were pseudonymously linked to the questionnaire responses using unique codes assigned by the enrolling centres. In cases of significant missing or implausible information in the medical record data, the study coordination centre (EBPI) contacted the respective clinical site to verify and clarify the entries. Similarly, when substantial data inconsistencies were identified in patient questionnaires, the enrolling centre was asked to follow up briefly with the patient for clarification.

Participating centres received a reimbursement per enrolled patient for whom complete data were submitted. Patients who completed the questionnaire were compensated with gift card (CHF50, i.e., about €52) as a sign of appreciation.

## Costs

The cost calculation was based on the resources used (e.g., PAH-specific treatment, outpatient visits, hospitalizations) and productivity losses (i.e., workdays lost) multiplied with corresponding unit costs and extrapolated to one year.

Hospitalization costs were estimated based on the length of stay in intensive care units (ICU), intermediate care, or general wards due to PAH. Due to the lack of PAH-specific hospitalization costs stratified by type of care, estimates from a report commissioned by the Swiss Confederation—based on data from four major health insurers (Helsana, SWICA,

Groupe Mutuel, and CSS) and focusing on COVID-19 hospitalizations—were used as a proxy. According to the report, average daily costs were €1,746 for general wards (range: €1,629–1,862) and €4,188 for ICUs (range: €3,723–4,653) [17]. For intermediate care stays, a midpoint value between general ward and ICU costs was assumed (€2,967 per day).

Outpatient visits at the participating centres were valued at €145.39 per visit, based on the Swiss outpatient medical tariff system (TARMED) [18]. Emergency department visits were assigned a cost of €480 per visit, as estimated by the Swiss Health Observatory [19]. Diagnostic tests and other procedures conducted during patient visits were costed separately. Only the 15 most frequently performed tests and procedures were included in the analysis, with detailed unit cost information provided in the supporting information (S1 Table in S1 File). Diagnostic tests that were rarely performed (e.g., nocturnal oximetry) were not considered in the analyses.

Costs for oxygen therapy (including equipment rental, oxygen refills, maintenance, and consumables) were obtained from the Swiss List of Means and Objects (MiGeL) and drug costs (per tablet or vial) from the Specialty List (Spezialitätenliste) [20,21].

For ambulatory consultations with general practitioners, costs were estimated based on a standard 30-minute consultation using the TARMED system, amounting to €106.43 per visit. This estimate did not include potential additional charges for diagnostic services, preparation of medical reports, or updates to patient records.

Specialist consultations outside the treating centres were assigned an average cost of €157 per visit. This estimate reflects the typical range of specialist consultation fees in Switzerland, which generally fall between €157 and €209 per visit [22,23].

Indirect costs were calculated using the human capital approach, which considers all hours lost due to illness and treatment—by both patients and informal caregivers—as productivity losses [24]. To estimate these costs over a four-week period, the reported times lost were multiplied by the approximate earnings of the individuals, derived from national statistics. For informal care, the proxy good method was applied. This approach assigns a monetary value to unpaid caregiving by equating it to the wage of a professional caregiver or equivalent substitute (e.g., a domestic worker) [25,26]. This method allows for monetization of services for which no direct financial transaction occurs. Specifically, we used age- and gender-specific average monthly wages from the Swiss Federal Statistical Office (SFSO) for the year 2020 [27]. These salaries reflect full-time employment (100% full-time equivalent). To convert monthly salaries to an hourly rate, we divided the figures by 173, in accordance with SFSO standards (based on a 40-hour work week over 4⅓ weeks per month). For employed patients, productivity losses were valued using the average wage of Swiss workers of matching age and gender. For patients not working due to PAH, we applied the same average wage over a four-week period. This value was adjusted based on the average level of employment of the working patients, estimated at 93% of a full-time equivalent. The value of time lost to household chores was based on estimates from a Swiss study, which set this at €28 per hour in 2022. Formal care was valued at €55.08 per hour, corresponding to the hourly cost of professional home care services (Spitex), which provide domestic and support services [28]. More details on unit costs used for the indirect cost calculation are provided in the supplementary materials (S2 Table in S1 File).

To calculate the economic burden at national level, the estimated total number of PAH patient in Switzerland was combined with the estimated total costs per patient. According to recently published estimates, the age-standardized prevalence of PAH in Europe ranges between 22.8 and 54.7 cases per million population. [9,10] Applying a mean of 38.8 cases per million to the Swiss population of 9 million yields an estimated 349 PAH cases nationwide (range: 205–492 cases). The total cost range reported in the results combine the prevalence range with the 95% confidence intervals of the total costs.

For international comparison, cost results are reported in Euro (€), using an average exchange rate of CHF1.00=€1.047 for the year 2024.

## Sample size

No formal sample size calculation was performed, as the study relied on descriptive statistics; however, based on clinical centre estimates, a target sample size of approximately 120 patients was anticipated.

## Statistical analysis

The demographic and clinical characteristics of the included population were described using counts and percentages for categorical variables, while continuous variables were described with means and standard deviations (SD). Chi-square tests and Fisher's exact tests were used for categorical variables. Kruskal-Wallis tests and Mann-Whitney U tests were used for continuous variables. To investigate the association between patient characteristics with total costs, generalized linear model (GLM) regressions with a log link function and assuming a gamma distribution of errors were used. Considering the uncertainty associated with annualizing costs derived from patient-reported outcomes collected over relatively short observation periods (4 weeks or 6 months), sensitivity analyses were conducted. For variables collected over 4 weeks and extrapolated to one year, multiplication factors ranging from 10 to 14 were used, with 13 applied in the main analysis. For variables collected over a 6-month period and extrapolated to one year, the multiplication factor used in the main analysis was varied by ±20%.

Missing data was handled using complete case analysis for each variable, without performing imputation. When participants provided qualitative responses to questions intended to collect quantitative data, these responses were classified as missing. The main database was prepared in Microsoft Excel. Statistical analyses were performed with IBM SPSS Statistics 29.0.2.0.

## Results

### Study participants

Between April and December 2024, a total of 177 patients diagnosed with PAH were contacted for study participation. Of these, 124 consented to participate and provided their personal data, yielding a participation rate of 70%. Approximately two-thirds of the participants were female, with a mean age of 62.1 years (median: 66 years). Non-responders had a mean age of 58.3 years (p = 0.162 for difference between responders and non-responders) and 59% were female (p = 0.393).

Given the statutory retirement age in Switzerland (65 years, excluding early retirement), over half of the cohort (53.2%) were retired. A total of 32 patients (25.8%) were classified as disabled (i.e., received money from the invalidity insurance), and only 17 (13.7%) were employed or self-employed at the time of the study.

Table 1 presents the distribution of participants including WHO-FC and ESC/ERS 2022 risk stratification, alongside data on comorbidities. The average time since diagnosis was 6.2 years. Patients in lower severity categories tended to have longer times since diagnosis. Specifically, mean ± SD time since PAH diagnosis by WHO-FC was 5.3 ± 4.6 years for Class 1, 7.3 ± 7.6 for Class 2, 5.4 ± 5.3 for Class 3, and 2.6 ± 2.7 for Class 4 (p = 0.284). Similarly, mean time since PAH diagnosis by ESC/ERS risk stratum was 7.1 ± 5.9 years for Stratum I (low risk), 6.0 ± 7.7 for Stratum II (intermediate-low risk), 5.1 ± 3.6 for Stratum III (intermediate-high risk), and 1.8 ± 1.0 for Stratum IV (high risk) (p = 0.167). For four patients, information on WHO-FC nor ESC/ERS risk was not available.

### Healthcare resources used

Table 2 summarizes healthcare resource utilization based on medical record data (covering 6 months) and patient-reported information (covering 4 weeks). Over the 6-month observation period, 23 participants (18.5%) experienced at least one PAH-related hospitalization. Among these, 15 patients were hospitalized once, 4 patients twice, another 4 patients three times, and 1 patient four times.

When averaged across the full study sample, the hospitalization rate was 0.29 ± 0.72 admissions per patient. The mean length of hospital stay was 2.55 ± 10.30 days. As shown in Table 2, patients classified in higher WHO-FC or ESC/ERS risk strata were hospitalized more frequently and had longer hospital stays.

The average number of outpatient visits at the participating centres over the 6-month period was 5.13 ± 5.34 per patient. Commonly performed diagnostic procedures during these visits included B-type natriuretic peptide (BNP) or N-terminal

**Table 1. Demographic and clinical characteristics of the study population.**

| Parameter, mean±SD or n (%) | Total respondents (N = 124) |
|---|---|
| Age (mean±SD) | 62.1±15.4 |
| Females | 81 (65.3%) |
| Education level | |
| *Compulsory* | 30 (24.4%) |
| *Upper secondary school* | 62 (50.4%) |
| *Tertiary education* | 31 (25.0%) |
| Employment status | |
| *Retired* | 66 (53.2%) |
| *Disable* | 32 (25.8%) |
| *Employed or self-employed* | 17 (13.7%) |
| *Others* | 9 (7.3%) |
| PAH subgroup | |
| *1.1 Idiopathic PAH* | 55 (44.4%) |
| *1.2 Heritable PAH* | 7 (5.6%) |
| *1.3 Drug- and toxin-inducted PAH* | 3 (2.4%) |
| *1.4.1 PAH associated with connective tissue disease* | 37 (29.8%) |
| *1.4.2 PAH associated with HIV infection* | 1 (0.8%) |
| *1.4.3 PAH associated with portal hypertension* | 7 (5.6%) |
| *1.4.4 PAH associated with congenital heart disease* | 8 (6.5%) |
| *1.5 PAH long-term responders to calcium channel blockers* | 0 (0) |
| *1.6 PAH with overt features of venous/capillaries involvement* | 6 (4.8%) |
| WHO functional class | |
| 1 | 26 (21.7%) |
| 2 | 52 (43.3%) |
| 3 | 37 (30.8%) |
| 4 | 5 (4.2%) |
| Risk stratification ESC/ERS | |
| I – Low risk | 48 (40.0%) |
| II – Intermediate-low risk | 43 (35.8%) |
| III – Intermediate-high risk | 25 (20.8%) |
| IV – High risk | 4 (3.3%) |
| Time since PAH diagnosis (mean±SD) | 6.2±6.2 |
| Number of co-morbidities per patient (mean±SD) | 3.4±2.1 |
| Main co-morbidities | |
| *Sleep Apnoea* | 43 (34.7%) |
| *Asthma* | 40 (32.3%) |
| *COPD* | 29 (23.4%) |
| *Pneumonia* | 8 (6.5%) |
| *Coronary artery disease* | 22 (17.7%) |
| *Hypertension* | 39 (31.7%) |
| *Other cardiovascular problems* | 35 (28.2%) |
| *Kidney disease* | 25 (20.2%) |
| *Diabetes* | 20 (16.1%) |
| *Systemic sclerosis* | 19 (15.3%) |
| *Cancer* | 17 (13.7%) |
| *Depression* | 15 (12.1%) |
| *Liver disease* | 13 (10.6%) |
| *Problems of the gastrointestinal tract* | 11 (8.9%) |
| *Osteoporosis* | 11 (8.9%) |
| *Rheumatoid arthritis* | 9 (7.3%) |

Abbreviations: COPD: Chronic obstructive pulmonary disease; ESC/ERS: European Society of Cardiology / European Respiratory Society; HIV: Human immunodeficiency virus; PAH: Pulmonary arterial hypertension SD: standard deviation.

**Table 2. Healthcare resource use according to medical record data and patient survey.**

| Parameter, mean±SD or n (%) | All (N=124) | By WHO functional class | | | | | By Risk stratification ESC/ERS* | | | | |
| --- | --- | --- | --- | --- | --- | --- | --- | --- | --- | --- | --- |
| | | 1 (N=26) | 2 (N=52) | 3 (N=37) | 4 (N=5) | p-value | I (N=48) | II (N=43) | III (N=25) | IV (N=4) | p-value |
| Number of hospitalizations (over 6 months) | 0.29±0.72 | 0.15±0.37 | 0.15±0.46 | 0.51±1.02 | 1.00±1.23 | 0.023 | 0.21±0.50 | 0.19±0.59 | 0.52±1.05 | 1.25±1.26 | 0.011 |
| Length of stay (days over 6 months) | 2.55±10.30 | 0.31±0.88 | 1.00±3.03 | 5.51±17.15 | 10.40±16.32 | 0.011 | 1.19±3.15 | 1.72±5.54 | 5.32±20.00 | 13.00±17.61 | 0.007 |
| ICU | 0 | 0 | 0 | 0 | 0 | – | 0 | 0 | 0 | 0 | – |
| Intermediate care | 0.14±0.92 | 0 | 0.06±0.42 | 0.35±1.60 | 0.20±0.45 | 0.124 | 0.06±0.43 | 0.09±0.61 | 0.36±1.80 | 0.25±0.50 | 0.119 |
| Normal ward | 2.43±9.61 | 0.27±0.78 | 0.94±2.94 | 5.24±15.79 | 10.20±16.38 | 0.015 | 1.10±3.05 | 1.72±5.54 | 4.92±18.24 | 12.75±17.73 | 0.008 |
| Number of outpatient visits to PAH centre (over 6 months) | 5.13±5.34 | 4.12±5.00 | 4.08±4.16 | 6.76±6.80 | 7.40±3.13 | 0.024 | 4.21±4.48 | 5.30±4.62 | 6.44±7.77 | 5.25±3.20 | 0.569 |
| Diagnostic tests (over 6 months) | | | | | | | | | | | |
| 6MWD | 1.07±0.94 | 1.16±0.85 | 0.98±0.76 | 1.24±1.21 | 0.40±0.55 | 0.236 | 1.09±0.76 | 1.21±1.15 | 1.00±0.87 | 0.25±0.50 | 0.204 |
| CPET | 0.19±0.45 | 0.24±0.52 | 0.18±0.43 | 0.22±0.80 | 0 | 0.717 | 0.28±0.50 | 0.16±0.48 | 0.12±0.33 | 0 | 0.211 |
| BNP or NT-proBNP | 1.59±1.41 | 1.20±0.71 | 1.59±1.80 | 1.76±1.64 | 2.40±1.32 | 0.100 | 1.35±0.90 | 1.56±1.24 | 2.00±2.25 | 2.25±1.50 | 0.344 |
| ECG | 0.48±0.85 | 0.36±0.76 | 0.39±0.72 | 0.57±0.93 | 1.60±1.34 | 0.044 | 0.37±0.77 | 0.44±0.88 | 0.68±0.85 | 1.00±1.41 | 0.167 |
| Echocardiography | 0.78±0.67 | 0.76±0.66 | 0.73±0.60 | 0.84±0.76 | 1.00±1.00 | 0.895 | 0.70±0.59 | 0.86±0.77 | 0.84±0.62 | 0.50±1.00 | 0.540 |
| cMRI | 0.08±0.28 | 0.04±0.20 | 0.04±0.20 | 0.16±0.37 | 0.20±0.48 | 0.129 | 0.07±0.25 | 0.14±0.35 | 0 | 0.25±0.50 | 0.135 |
| Arterial blood gas or pulse oximetry | 0.98±1.32 | 0.76±0.93 | 0.65±0.77 | 1.54±1.87 | 2.00±1.58 | 0.013 | 0.80±0.91 | 1.16±1.76 | 0.92±1.00 | 2.25±1.17 | 0.317 |
| Hemodynamics (e.g., doppler ultrasound) | 0.55±1.20 | 0.32±0.69 | 0.67±1.62 | 0.32±0.48 | 1.20±1.10 | 0.200 | 0.33±0.52 | 0.51±1.32 | 0.76±1.69 | 1.00±1.15 | 0.551 |
| Right Hearth Catheterization | 0.34±0.48 | 0.23±0.43 | 0.33±0.47 | 0.32±0.48 | 0.60±0.55 | 0.431 | 0.29±0.46 | 0.30±0.46 | 0.36±0.49 | 0.50±0.58 | 0.799 |
| Pulmonary function test | 0.86±0.76 | 1.08±0.86 | 0.67±0.59 | 1.03±0.87 | 0.80±0.84 | 0.132 | 0.84±0.69 | 0.86±0.91 | 1.00±0.65 | 0.25±0.50 | 0.199 |
| CT scan | 0.24±0.55 | 0.40±0.76 | 0.20±0.53 | 0.16±0.37 | 0.60±0.55 | 0.089 | 0.28±0.69 | 0.21±0.41 | 0.25±0.53 | 0.25±0.50 | 0.997 |
| Blood/Urine test | 0.70±1.47 | 0.28±0.74 | 0.27±0.79 | 1.68±2.21 | 0.60±0.89 | <0.001 | 0.43±0.93 | 0.98±1.88 | 0.88±1.59 | 0.50±1.00 | 0.365 |
| Radiography | 0.10±0.39 | 0 | 0.06±0.24 | 0.16±0.44 | 0.60±1.34 | 0.141 | 0.04±0.20 | 0.07±0.34 | 0.28±0.68 | 0 | 0.070 |
| Sonography | 0.10±0.35 | 0.00 | 0.06±0.24 | 0.24±0.55 | 0.60±1.34 | 0.035 | 0.06±0.25 | 0.14±0.47 | 0.12±0.33 | 0 | 0.782 |
| Treprostinil pump filling | 1.18±3.14 | 0.68±2.39 | 1.17±3.35 | 1.32±3.15 | 0 | 0.406 | 1.30±3.56 | 1.02±2.84 | 0.80±2.48 | 0.50±1.00 | 0.976 |
| Number of emergency visits (over 6 months) | 0.11±0.39 | 0.12±0.33 | 0.04±0.19 | 0.11±0.32 | 1.00±1.23 | | 0.10±0.31 | 0.02±0.15 | 0.12±0.33 | 1.25±1.26 | |
| Proportion of patients with PAH treatment (in the last 6 months) | | | | | | | | | | | |
| Longterm oxygen therapy | 46 (37.1%) | 2 (7.7%) | 18 (34.6%) | 20 (54.1%) | 5 (100%) | <0.001 | 11 (22.9%) | 19 (44.2%) | 11 (44.0%) | 4 (100%) | 0.006 |
| Macitentan (p.o.) | 87 (70.2%) | 19 (73.1%) | 33 (63.5%) | 31 (83.8%) | 3 (60.0%) | 0.190 | 37 (77.1%) | 30 (69.8%) | 16 (64.0%) | 3 (75.0%) | 0.679 |
| Tadalafil (p.o) | 57 (46.0%) | 13 (50.0%) | 19 (36.5%) | 23 (62.3%) | 1 (20.0%) | 0.064 | 20 (41.7%) | 21 (48.8%) | 12 (48.0%) | 2 (50.0%) | 0.904 |

Table 2. (Continued)

| Parameter, mean±SD or n (%) | All (N=124) | By WHO functional class | | | | | By Risk stratification ESC/ERS* | | | | |
|---|---|---|---|---|---|---|---|---|---|---|---|
| | | 1 (N=26) | 2 (N=52) | 3 (N=37) | 4 (N=5) | p-value | I (N=48) | II (N=43) | III (N=25) | IV (N=4) | p-value |
| Riociguat (p.o.) | 21 (16.9%) | 2 (7.7%) | 13 (25.0%) | 4 (10.8%) | 1 (20.0%) | 0.167 | 8 (16.7%) | 11 (25.6%) | 1 (4.0%) | 1 (25.0%) | 0.152 |
| Treprostinil (i.v.) | 20 (16.1%) | 2 (7.7%) | 8 (15.4%) | 8 (21.6%) | 0 | 0.355 | 7 (14.6%) | 6 (14.0%) | 4 (16.0%) | 1 (25.0%) | 0.945 |
| Sildenafil (p.o.) | 17 (13.7%) | 2 (7.7%) | 6 (11.5%) | 6 (16.2%) | 2 (40%) | 0.240 | 4 (8.3%) | 4 (9.3%) | 7 (28.0%) | 1 (25.0%) | 0.080 |
| Selexipag (p.o.) | 14 (11.3%) | 1 (3.8%) | 10 (19.2%) | 3 (8.1%) | 0 | 0.136 | 7 (14.6%) | 6 (14.0%) | 1 (4.0%) | 0 | 0.463 |
| Ambrisentan (p.o.) | 10 (8.1%) | 1 (3.8%) | 6 (11.5%) | 2 (5.4%) | 1 (20.0%) | 0.442 | 2 (4.2%) | 5 (11.6%) | 2 (8.0%) | 1 (25.0%) | 0.368 |
| Bosentan (p.o.) | 8 (7.3%) | 1 (3.8%) | 6 (11.5%) | 0 | 1 (20.0%) | 0.094 | 2 (4.2%) | 3 (7.0%) | 3 (12.0%) | 0 | 0.590 |
| Torasemid (p.o.) | 6 (4.8%) | 0 | 4 (7.7%) | 2 (5.4%) | 0 | 0.487 | 0 | 3 (7.0%) | 3 (12.0%) | 0 | 0.129 |
| Amlodipin (p.o.) | 3 (2.4%) | 0 | 1 (1.9%) | 1 (2.7%) | 0 | 0.852 | 1 (2.1%) | 0 | 1 (4.0%) | 0 | 0.642 |
| Number of visits outside PAH centres (over 4 weeks) | | | | | | | | | | | |
| Number of visits to family doctor | 0.59±0.92 | 0.38±0.70 | 0.56±0.92 | 0.70±1.08 | 1.50±0.58 | 0.048 | 0.44±0.80 | 0.70±1.01 | 0.67±1.00 | 1.25±0.96 | 0.159 |
| Number of visits to specialists | 1.51±2.86 | 1.04±2.01 | 1.33±3.24 | 1.92±2.75 | 2.75±2.50 | 0.129 | 0.92±2.07 | 2.23±3.68 | 1.41±2.60 | 3.00±2.16 | 0.010 |
| Formal care hours per week | 0.67±1.91 | 0.56±1.71 | 0.20±1.15 | 0.99±1.75 | 2.40±4.34 | 0.005 | 0.47±1.69 | 0.30±1.12 | 1.33±2.46 | 1.25±1.50 | 0.012 |
| Productivity loss | | | | | | | | | | | |
| Work hours lost (over 4 weeks) | 30.70±59.14 | 19.95±51.96 | 36.71±63.24 | 29.24±57.56 | 29.76±66.55 | 0.549 | 46.97±68.96 | 22.93±52.16 | 10.75±37.45 | 37.20±74.40 | 0.019 |
| Household chores, hours lost per week | 8.49±34.78 | 1.79±3.74 | 12.27±50.81 | 5.95±9.58 | 29.00±29.92 | 0.063 | 12.01±53.84 | 6.58±12.78 | 5.00±10.70 | 20.67±34.08 | 0.676 |
| Informal care hours per week | 4.71±15.62 | 1.20±2.90 | 6.81±21.55 | 2.44±4.49 | 17.25±28.65 | 0.092 | 6.98±22.49 | 2.53±4.54 | 2.67±4.59 | 20.00±34.64 | 0.957 |
| Informal care by a caregiver below retirement age per week | 3.39±13.77 | 1.15±2.85 | 5.37±20.35 | 1.97±6.29 | 4.00±6.76 | 0.521 | 6.13±20.94 | 2.15±6.03 | 0.57±1.31 | 4.67±8.08 | 0.730 |

Abbreviations: 6MWD: 6-minutes walking distance; BNP: Brain natriuretic peptide; cMRI: Cardiac magnetic resonance imaging"; CPET: Cardiopulmonary exercise testing; CT: Computed tomography; ECG: Electrocardiogram; ESC/ERS: European Society of Cardiology / European Respiratory Society; HIV: Human immunodeficiency virus; i.v.: intravenous; ICU: Intensive Care Unit; NT-proBNP: N-terminal pro-B-type natriuretic peptide; p.o.: per os (orally), PAH: Pulmonary arterial hypertension SD: Standard deviation.

*I = low risk, II = intermediate-low risk, III = intermediate-high risk, IV = high risk.

pro-BNP (NT-proBNP) testing, 6-minute walk distance (6MWD) assessments, arterial blood gas analysis or pulse oximetry, and pulmonary function tests. The frequency of diagnostic testing tended to increase with worsening WHO-FC or risk category (Table 2).

Longterm oxygen therapy was administered to 45 patients (38%), with the proportion of patients requiring oxygen increasing markedly in WHO-FC 4 and ESC/ERS risk IV.

Regarding pharmacological treatment, the most frequently prescribed medication was macitentan (N = 46; 70%), followed by tadalafil (N = 57; 46%), riociguat (N = 21; 17%), treprostinil (N = 20; 16%), sildenafil (N = 17; 14%), selexipag (N = 14; 11%), and ambrisentan (N = 10; 8%). Other PAH-specific therapies were used in fewer than 7% of patients. The number of PAH drugs per patient ranged from 0 (N = 7, 6%) to 4 (N = 3, 2%). Most patients received at least 2 (N = 63, 51%) or 3 (N = 29, 23%) PAH drugs simultaneously.

In addition to outpatient visits at their treating PAH centres, patients reported an average of 0.60 ± 0.93 visits to a general practitioner and 1.50 ± 2.83 visits to medical specialists over the 4 weeks preceding study enrolment. The mean number of hours of formal care received per patient was 0.6 ± 1.7 hours during the same 4-week period.

Regarding productivity losses, the average number of work hours lost per patient—composed of losses due to absence from work and disability, and averaged across all study participants—was 28.1 ± 57.1 hours over a 4-week period. This corresponds to an estimated loss of approximately five full working days per patient per month. The mean number of hours of informal care averaged 4.7 ± 15.8 hours per patient, of which approximately 70% (3.3 hours) was provided by non-retired caregivers.

More details on the distribution and uncertainty of the collected data (e.g., in terms of median, minimum, maximum, and 95% confidence intervals) are provided in the supplementary materials (S3 Table in S1 File).

## Estimated annual costs per patient

Table 3 presents the estimated annual direct and indirect costs per PAH patient, both for the overall study population and stratified by WHO-FC and ESC/ERS risk category (Figs 1 and 2). The mean total cost per patient was estimated at

**Table 3. Estimated direct and indirect costs per patient per year (EUR).**

| | | By WHO functional class | | | | | By Risk stratification ESC/ESR* | | | | |
|---|---|---|---|---|---|---|---|---|---|---|---|
| | All (N = 124) | 1 (N = 26) | 2 (N = 52) | 3 (N = 37) | 4 (N = 5) | p-value | I (N = 48) | II (N = 43) | III (N = 25) | IV (N = 4) | p-value |
| Diagnostic costs | 2,359 | 1,989 | 1,920 | 3,109 | 3,365 | 0.024 | 2,050 | 2,603 | 2,522 | 2,527 | 0.569 |
| Outpatient visit to PAH centre costs | 1,491 | 1,197 | 1,185 | 1,964 | 2,152 | 0.021 | 1,224 | 1,541 | 1,872 | 1,527 | 0.441 |
| Treatment costs | 90,205 | 58,176 | 92,918 | 100,391 | 54,717 | 0.063 | 78,204 | 89,299 | 82,990 | 180,499 | 0.214 |
| Hospitalization costs | 9,288 | 940 | 3,632 | 20,389 | 36,795 | 0.016 | 4,225 | 6,560 | 19,312 | 45,994 | 0.009 |
| GP visit costs | 815 | 532 | 772 | 972 | 1,660 | 0.159 | 605 | 965 | 886 | 1,729 | 0.163 |
| Specialist visits costs | 3,063 | 2,120 | 2,710 | 3,918 | 4,492 | 0.228 | 1,872 | 4,558 | 2,777 | 6,125 | 0.010 |
| Formal care costs | 1,894 | 1,597 | 551 | 2,748 | 6,873 | 0.005 | 1,342 | 832 | 3,666 | 3,580 | 0.014 |
| **Total direct costs** | **109,114** | **66,552** | **103,687** | **133,492** | **110,055** | **0.012** | **89,522** | **106,361** | **114,026** | **241,980** | **0.064** |
| Workdays lost costs | 16,944 | 10,553 | 20,366 | 16,108 | 15,695 | 0.486 | 25,875 | 12,415 | 6,144 | 19,618 | 0.019 |
| Household chores costs | 7,469 | 2,431 | 8,440 | 6,854 | 34,107 | 0.196 | 6,569 | 8,325 | 6,469 | 22,787 | 0.758 |
| Informal care costs | 5,431 | 2,421 | 7,342 | 3,912 | 6,713 | 0.697 | 9,004 | 4,293 | 1,091 | 7,342 | 0.772 |
| **Total indirect costs** | **29,844** | **15,405** | **36,148** | **26,875** | **56,515** | **0.219** | **41,447** | **25,033** | **13,704** | **49,748** | **0.318** |
| **Total costs** | **138,959** | **81,957** | **139,836** | **160,366** | **166,569** | **0.011** | **130,970** | **131,393** | **127,730** | **291,728** | **0.083** |

*I = low risk, II = intermediate-low risk, III = intermediate-high risk, IV = high risk.

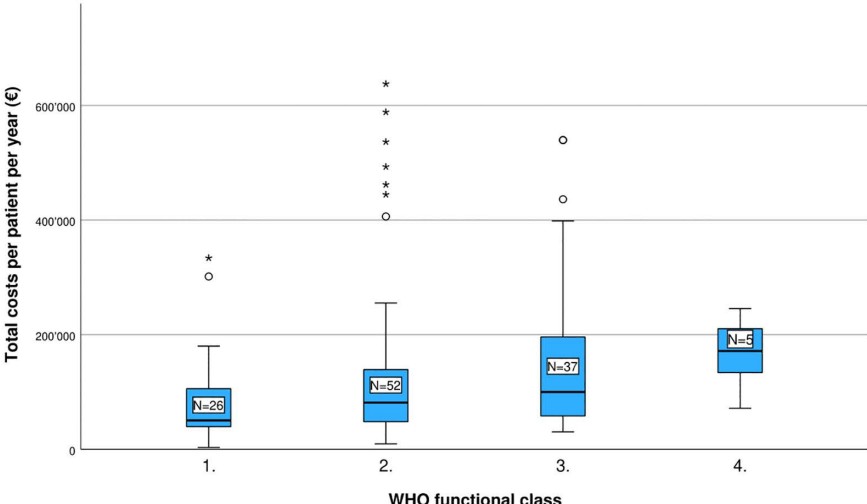

**Fig 1. Estimated total costs of PAH per patient per year (EUR), by WHO functional class.**

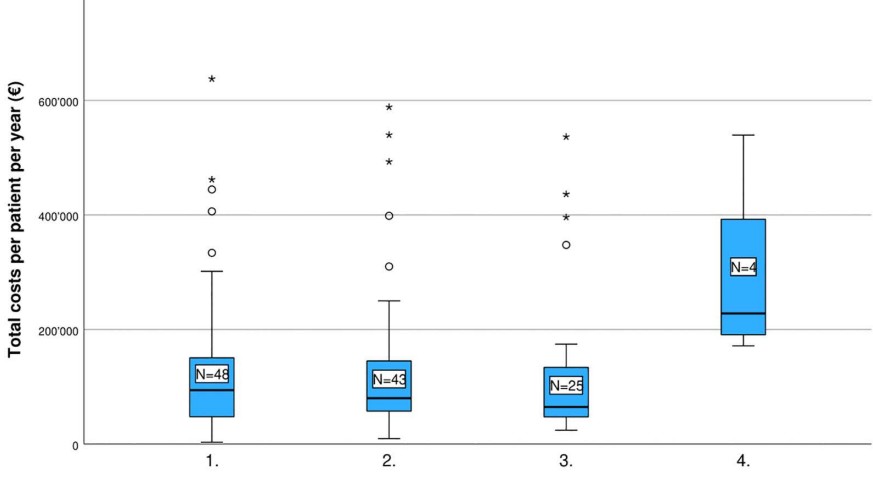

**Fig 2. Estimated total costs of PAH per patient per year (EUR), by Risk stratification ESC/ERS.**

€138,958 per year. Of this, direct costs accounted for approximately 79%, while indirect costs represented the remaining 21%.

Pharmacological treatment emerged as the primary cost component, comprising 65% of the total annual costs and 83% of direct healthcare costs. In contrast, costs related to hospitalizations, diagnostics, and outpatient visits contributed only a small proportion to the overall economic burden. More details on the distribution and uncertainty of the cost estimates (e.g., in terms of median, minimum, maximum, and 95% confidence intervals) are provided in the supplementary materials (S4 Table in S1 File).

When stratified by WHO-FC, average total annual costs increased with disease severity: €81,957 for Class 1, €139,836 for Class 2, €160,366 for Class 3, and €166,569 for Class 4 (p = 0.011). A similar pattern was observed when stratifying by

ESC/ERS risk categories, though with some variation: €130,970 for low risk, €131,393 for intermediate-low risk, €127,730 for intermediate-high risk, and €291,728 for high risk (p = 0.083). To assess the robustness of cost estimates across disease severity groups, additional distributional analyses were conducted (see S5 Table in S1 File). Generally, greater variability and wider confidence intervals were observed in higher-risk groups, reflecting small subgroup sizes and the influence of high-cost cases.

Exploratory analyses indicated that treatment with intravenous Treprostinil was a major cost driver. Patients receiving Treprostinil (n = 20) incurred substantially higher costs compared to those not receiving it. The mean annual direct costs for Treprostinil-treated patients were €350,967, compared to €62,604 for those not on this therapy. Similarly, indirect costs were markedly higher in this group, averaging €54,370 versus €25,128. Among the patients who did not receive Treprostinil, the costs increased with disease severity: €62,312 for Class 1, €81,476 for Class 2, €104,547 for Class 3, and €166,570 for Class 4 (p = 0.006). A different pattern was observed when stratifying by ESC/ERS risk categories: €87,377 for low risk, €84,444 for intermediate-low risk, €82,794 for intermediate-high risk, and €209,166 for high risk (p = 0.046).

Another exploratory subgroup analysis was conducted among patients with connective tissue disease (CTD)–associated PAH to compare systemic sclerosis (SSc) and non-SSc cases. Of the 37 patients with CTD-associated PAH, 19 had SSc. Patients with SSc were more frequently female (84.2% vs. 61.1%) and older (mean age 69.8 vs. 62.4 years) compared with non-SSc patients. Mean annual direct and indirect costs were higher in non-SSc patients (€139,080 and €44,153, respectively) than in those with SSc (€64,981 and €25,985, respectively).

In a further analysis, patients were stratified according to the time since PAH diagnosis. If compared to prevalent cases (i.e., those diagnosed more than 1 year age), incident cases generally had higher costs related to diagnostic and outpatient visits to PAH centres, but considerably lower costs related to PAH treatment (see S6 Table in S1 File).

In the first GLM regression with total costs as dependent variable, males, ESC/ERS Risk Stratum IV and disease duration were significantly associated with increased costs. On the other hand, females, age, and being employed/self-employed were associated with decreasing costs (see S7 Table in S1 File). The second GLM aimed to investigate which factors among WHO-FC, ESC/ERS 2022 risk strata, and PAH clinical subgroup classification had the highest impact on total costs (see S8 Table in S1 File). Treatment with treprostinil was the strongest determinant of total costs (coefficient 1.432, p < 0.001). Higher WHO-FC was also significantly associated with increased costs, with the largest effect observed for WHO-FC 4 (coefficient 0.970, p = 0.035), followed by WHO-FC 3 (coefficient 0.745, p < 0.001) and WHO-FC 2 (coefficient 0.379, p = 0.022). In contrast, PAH clinical subtype and ESC/ERS 2022 risk strata were not significantly associated with total healthcare costs in the multivariable model. It is important to emphasize that the regression analysis was conducted in a relatively small sample, which may limit statistical power and the stability of model estimates. In addition, several covariates included in the model, particularly WHO-FC and ESC/ERS risk strata, partially overlap conceptually and clinically, as both reflect disease severity. WHO-FC and ESC/ERS risk strata were in fact significantly correlated (correlation coefficient 0.575, p < 0.001). This collinearity may have attenuated independent associations and should be considered when interpreting the relative contribution of individual predictors to healthcare costs.

In the sensitivity analysis, in which outcomes collected over 4 weeks and 6 months were annualized using lower and higher multiplication factors, estimated direct costs ranged from €87,114 to €130,227, indirect costs from €22,957 to €32,140, and total costs from €110,071 to €162,367. In general, the relative distribution of the costs components remained stable. Varying the multiplication factor for outcomes collected over a 4-week period had little impact on cost estimates. In contrast, varying the annualization of variables collected over a 6-month period (particularly treatment costs) had a moderate impact on the estimated costs (see S9 Table in S1 File). Graphical representations of additional subgroup analyses (e.g., by gender, age group, age since PAH diagnosis, PAH subtype, number of co-morbidities, and number of prescribed PAH drugs) are provided in the supporting information (Figs S1 to S15 in S1 File).

## Estimated national burden

Assuming the study sample is representative of the broader PAH population in Switzerland with respect to healthcare utilization and productivity losses, the total annual economic burden of PAH is projected at €48.5 million (range: €23.2–81.0 million). Of this, €37.0 million (range: €17.7–64.8 million) would be attributable to direct costs, and €12.5 million (range: €4.4–18.9 million) to indirect costs.

## Discussion

This study provides the first comprehensive estimate of the economic burden of PAH in Switzerland, capturing both direct and indirect costs using real-world data from medical records and patient-reported outcomes. The mean annual cost per PAH patient was estimated at more than €138,000, with an extrapolated national burden of approximately €48.5 million.

More than half of the direct healthcare costs was for pharmacological treatment of PAH. This finding is consistent with international literature, where PAH-specific therapies—often high-cost and lifelong—are recognized as key cost drivers [29]. Conversely, costs associated with hospitalizations, diagnostics, and outpatient visits played a comparatively minor role, though they may increase with disease severity.

Indirect costs, primarily driven by lost work productivity, contributed moderately to the total burden. The average patient lost approximately five working days per month, and an important proportion of the received informal care was provided by caregivers below the age of retirement. These figures underscore the societal impact of PAH beyond the healthcare system, affecting employment, caregivers, and overall economic productivity.

As expected, costs escalated substantially with disease severity, both when stratified by WHO-FC and ESC/ERS risk strata. The observed increase in total annual costs with worsening WHO-FC (from €81,957 for WHO-FC 1, €139,836 for WHO-FC 2, €160,366 for WHO-FC 3, to €166,569 for WHO-FC 4) is consistent with the broader literature on the economic burden of PAH. The systematic literature review by Ramani et al. identified a similar pattern across multiple geographies and healthcare settings, with four independent studies all reporting a general increase in total healthcare costs for patients in higher WHO-FC groups [30]. In the US study by Dufour et al., mean yearly total costs increased from $73,443 for WHO-FC 1 to $153,732 for WHO-FC 2, $145,230 for WHO-FC 3, and $175,368 for WHO-FC 4 (converted to 2024 USD), suggesting that the most pronounced cost step-up occurs between WHO-FC 1 and WHO-FC 2, with more modest incremental increases thereafter [31]. This pattern mirrors the gradient observed in our study. In the Spanish study by ZoZaya et al., mean annual total costs among prevalent PAH patients were reported as $30,427 for WHO-FC 1–2, $52,952 for WHO-FC 3, and $112,845 for WHO-FC 4, while for incident patients the corresponding figures were $77,333, $122,978, and $247,556, respectively, indicating a steeper cost escalation in the most advanced disease stages than observed in our cohort [32]. Similarly, the study by Tassara et al., conducted in Argentina, reported annual cost ranges of $17,855–$28,616 for WHO-FC 1, $59,437–$80,542 for WHO-FC 2, $198,456–$284,320 for WHO-FC 3, and $378,894–$555,043 for WHO-FC 4, reflecting a markedly steeper gradient than observed in our study [33]. In the Australian PAH-systemic sclerosis cohort studied by Morrisroe et al., the cost gradient was less pronounced, with mean annual healthcare costs of $6,432 for WHO-FC 1, $6,338 for WHO-FC 2, $9,761 for WHO-FC 3, and $8,966 for WHO-FC 4 [34]. The lower costs compared to the previously mentioned studies is presumably due to the fact that this assessment did included hospitalisation and ambulant care costs, but excluded medication costs. Taken together, these data confirm that disease severity as measured by WHO-FC is a consistent and significant determinant of economic burden in PAH across diverse healthcare systems, and that preventing progression to higher WHO-FC groups through timely diagnosis and effective treatment escalation represents a possible pathway to reduce the overall economic burden of the disease.

To our knowledge, this study is the first to report total healthcare costs stratified by the ESC/ERS risk classification in patients with PAH. Our findings, showing total annual costs of €130,970, €131,393, €127,730, and €291,728 for low, intermediate-low, intermediate-high-, and high-risk patients respectively, represent a novel contribution to the economic literature on PAH and underscore the cost burden associated with high-risk disease. The lack of statistical significance,

alongside the relatively flat cost gradient across the three lower risk categories, may partly reflect limited sample size in the higher risk groups. Additional studies with larger cohorts are needed to fully characterize the relationship between ESC/ERS risk category and healthcare costs in PAH.

The findings of this study carry several important clinical implications for the future management of PAH. The clear association between worsening WHO-FC and increasing total annual healthcare costs provides a strong economic rationale for aggressive early intervention. Keeping patients at WHO-FC 1 or WHO-FC 2, rather than allowing progression to WHO-FC 3 or WHO-FC 4, could translate into substantial cost savings while simultaneously reducing the broader burden experienced by patients with more advanced disease. This is consistent with Ramani et al. (2025), who suggested that treatment escalation costs can be offset by meaningful reductions in healthcare resource utilization. The considerable cost burden identified in high-risk patients according to the ESC/ERS risk classification — with total annual costs of €291,728 in high-risk patients compared with approximately €130,000 across the three lower risk categories — further highlights the economic penalty of failing to achieve guideline-recommended low-risk status. From a healthcare system perspective, investment in proactive monitoring, multidisciplinary care, and timely treatment escalation is therefore likely to be economically justified if it succeeds in keeping a greater proportion of patients within lower risk categories.

These findings also have implications for how the value of newer PAH therapies should be assessed. The economic value of agents capable of achieving meaningful risk downgrading — including recently approved treatments such as sotatercept — should be evaluated in the context of total cost of care rather than pharmacy costs alone, as fewer hospitalizations associated with improved risk status may offset incremental drug acquisition costs. At the same time, whether the net economic impact of such agents is favourable remains uncertain given their high prices. More broadly, the concentration of costs in higher WHO-FC and high-risk groups reflects not only greater clinical complexity but also greater patient burden, including reduced functional independence and impaired ability to work. Reducing disease progression through optimal management therefore has the potential to meaningfully improve quality of life and preserve productivity — and the extent to which novel therapies enable meaningful workforce re-entry of working-age patients warrants further investigation [35,36]. Future studies should examine the real-world impact of novel therapies on the full spectrum of direct and indirect costs, including informal care and productivity losses, alongside quality-of-life outcomes. The present work provides an important baseline against which the future impact of evolving treatment strategies can be assessed.

## Strengths and limitations

The study has several strengths, including the use of detailed, patient-level clinical and cost data from multiple Swiss centres, and the adoption of a societal perspective in cost estimation. Indirect costs were assessed through a validated, simple, and short questionnaire. Although the survey was not pilot tested, the participation rate was high and there were very few missing or implausible values, implying that the questionnaire was well accepted and understood by the study participants. However, several limitations should be noted. First, although we could include a substantial proportion of the Swiss PAH population, the absolute number of included PAH patients remained small due to the rarity of the condition, implying a risk of residual chance effects. Second, considerable variability was observed across multiple variables (e.g., length of stay, number of diagnostic tests, costs). This heterogeneity mainly reflected the limited sample size and substantial inter-patient differences in health care utilization. Third, to calculate the national burden of PAH, we applied published prevalence estimates ranging from 22.8 to 54.7 PAH cases per million in the absence of Swiss-specific data. According to clinical experts, the real number of PAH patients in Switzerland likely approximates the upper bound of this range. In the future, data from the Swiss pulmonary hypertension registry may enable a more accurate estimation. Fourth, some cost variables were based on outcomes collected over relatively short observation periods. Patient-reported outcomes were collected over 4 weeks, while other outcomes were extracted from medical records covering a 6-month period. Extrapolating these variables to one year by simple multiplication implies the assumption that the impact of PAH on patient health and resource use remains constant over time. However, seasonal effects, which could lead, for example, to more frequent exacerbations, cannot be excluded. The sensitivity analysis conducted to assess

the uncertainty related to the annualization of costs showed that varying the multiplication factor for outcomes collected over 4 weeks (10–14 instead of 13 used in the main analysis) had a minimal impact on direct costs and only a small impact on indirect costs. For variables collected over a 6-month period, the multiplication factor used in the main analysis was varied by ±20% in the sensitivity analysis; this resulted in a moderate impact on cost estimates, particularly for treatment costs. Fifth, we cannot exclude non-response bias, also given that non-responders had a lower mean age and were more often male than responders (both age and gender were not statistically different between responders and non-responders). This may have led to non-representative results for healthcare resource consumption and productivity losses. Sixth, we are not able to verify self-reported loss of time and healthcare use outside the enrolling hospital: although we adhered to the timeframe specified by the iPCQ, some patients may have encountered difficulties recalling the exact amount of time lost or the number of visits in the last 4 weeks. Seventh, this study focused exclusively on medications indicated for PAH and did not account for treatments related to concomitant diseases. Given that most participants were polymorbid, some healthcare resource use and productivity losses may have been attributable to comorbidities rather than PAH itself. Consequently, the reported costs may both underestimate the true economic burden—by excluding therapies required for associated conditions (e.g., steroids or immunosuppressants in patients with connective tissue disease)—and overestimate PAH-specific costs by capturing resource utilization driven by coexisting diseases. Eighth, when patient recruitment started, three principles pathways for treatment of PAH were available (endothelin, nitric oxide, prostacyclin). Newer therapies such as activin-signalling inhibitors were not yet approved/available and their impact is therefore not considered in this article. Ninth, among the participating PAH patients, only one individual received epoprostenol in combination with macitentan, prescribed in the context of a late PAH diagnosis during pregnancy, where it represented the most appropriate therapeutic option. Epoprostenol is not included in the list of medications covered by mandatory basic health insurance in Switzerland, and no official Swiss public price is available. One option would have been to directly contact the pharmaceutical company to obtain the applicable cost. However, given that this concerns a single patient in our cohort, doing so could raise ethical concerns regarding patient rights. Therefore, for this patient, only macitentan costs were included in the analysis, while potential epoprostenol costs were omitted. Given that this concerns a single case, we expect this omission to have a limited impact on the overall cost estimates. Finally, cross-sectional data cannot capture longitudinal cost dynamics or treatment trajectories.

Despite these limitations, the findings underscore the high financial impact of PAH in Switzerland and highlight the importance of optimizing disease management and resource allocation for this rare, high-cost condition.

## Conclusion

Pulmonary arterial hypertension imposes a substantial economic burden in Switzerland, with average costs per patient distinctly above €100,000 per year. Direct medical costs, particularly those related to pharmacological treatment, dominate overall, while productivity losses contribute significantly to indirect costs. Extrapolated to the national level, the total annual burden is estimated at €48.5 million. These findings highlight the urgent need for strategies that not only improve clinical outcomes but also reduce the broader societal and economic impact of PAH.

## Supporting information

**S1 File. Supporting information.** Additional tables and figures.
(DOCX)

## Acknowledgments

We sincerely thank all PAH patients who participated in this study for their valuable contributions. We also gratefully acknowledge the dedicated nurses at the participating clinical centres for their support in extracting the medical record data.

## Author contributions

**Conceptualization:** Yuki Tomonaga, Sandro Stoffel, Matthias Schwenkglenks.

**Data curation:** Yuki Tomonaga, Mona Lichtblau, Silvia Ulrich, Sabina A Guler, Patrick Yerly, Benoît Lechartier, Jean-Marc Fellrath, Anne Bergeron, Louise Bondeelle, Silviu-Mihail Chirila.

**Formal analysis:** Yuki Tomonaga.

**Funding acquisition:** Andrea Favre-Bulle.

**Investigation:** Yuki Tomonaga, Mona Lichtblau, Silvia Ulrich, Sabina A Guler, Patrick Yerly, Benoît Lechartier, Anne Bergeron, Louise Bondeelle, Silviu-Mihail Chirila.

**Methodology:** Yuki Tomonaga, Sandro Stoffel, Matthias Schwenkglenks.

**Project administration:** Yuki Tomonaga.

**Software:** Yuki Tomonaga.

**Supervision:** Matthias Schwenkglenks.

**Validation:** Matthias Schwenkglenks.

**Visualization:** Yuki Tomonaga, Sandro Stoffel.

**Writing – original draft:** Yuki Tomonaga, Sandro Stoffel, Matthias Schwenkglenks.

**Writing – review & editing:** Yuki Tomonaga, Mona Lichtblau, Silvia Ulrich, Sabina A Guler, Patrick Yerly, Benoît Lechartier, Jean-Marc Fellrath, Anne Bergeron, Louise Bondeelle, Silviu-Mihail Chirila, Andrea Favre-Bulle, Sandro Stoffel, Matthias Schwenkglenks.

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
