## [Decision Letter · Decision Letter 0]

19 Feb 2026

PONE-D-25-57450Economic Burden of Pulmonary Arterial Hypertension in Switzerland.PLOS One

Dear Dr. Tomonaga,

Thank you for submitting your manuscript to PLOS ONE. After careful consideration, we feel that it has merit but does not fully meet PLOS ONE’s publication criteria as it currently stands. Therefore, we invite you to submit a revised version of the manuscript that addresses the points raised during the review process.

We look forward to receiving your revised manuscript.

Kind regards,

Yoshihiro Fukumoto

Academic Editor

PLOS One

**Journal Requirements:**

1. When submitting your revision, we need you to address these additional requirements. Please ensure that your manuscript meets PLOS ONE's style requirements, including those for file naming. The PLOS ONE style templates can be found at https://journals.plos.org/plosone/s/file?id=wjVg/PLOSOne_formatting_sample_main_body.pdf and https://journals.plos.org/plosone/s/file?id=ba62/PLOSOne_formatting_sample_title_authors_affiliations.pdf 2. We note that the grant information you provided in the ‘Funding Information’ and ‘Financial Disclosure’ sections do not match.  When you resubmit, please ensure that you provide the correct grant numbers for the awards you received for your study in the ‘Funding Information’ section. 3. Please expand the acronym “YT, SS, MS, ML, SU, SG, PY, BL, J-MF, AB, LB and S-MC” (as indicated in your financial disclosure) so that it states the name of your funders in full.This information should be included in your cover letter; we will change the online submission form on your behalf. 4. Thank you for stating the following in the Competing Interests section: YT, SS, and MS received financial support from MSD through their employment institutions for organizing and conducting the study, analyzing the collected data, and writing the report/manuscript. ML reports grants, honoraria, or consulting fee from MSD, Johnson&Johnson, Gebro Pharma, and Orpha Swiss. SU reports research grants, honoraria, or consulting fee from the Swiss National Science Foundation, Zurich and Swiss Lung League and EMDO foundation, Orpha Swiss, Janssen SA, MSD SA, Gebro SA, Ideogen and Astra Zeneca (all unrelated to the present work). SG reports grants, honoraria, or consulting fee from MSD, Johnson&Johnson, Gebro Pharma, and Orpha Swiss. AB reports grants, honoraria, or consulting fee from AstraZeneca, Sanofi, GSK, Novartis, OM pharma, Boehringer Ingelheim. MS reports grants, honoraria, or consulting fee from, AbbVie, Bristol-Myers Squibb, Novartis, Pfizer, Roche, and AstraZeneca. Other authors declare no conflicts of interest in relation to this manuscript.  We note that one or more of the authors are employed by a commercial company.  a. Please provide an amended Funding Statement declaring this commercial affiliation, as well as a statement regarding the Role of Funders in your study. If the funding organization did not play a role in the study design, data collection and analysis, decision to publish, or preparation of the manuscript and only provided financial support in the form of authors' salaries and/or research materials, please review your statements relating to the author contributions, and ensure you have specifically and accurately indicated the role(s) that these authors had in your study. You can update author roles in the Author Contributions section of the online submission form. Please also include the following statement within your amended Funding Statement. “The funder provided support in the form of salaries for authors, but did not have any additional role in the study design, data collection and analysis, decision to publish, or preparation of the manuscript. The specific roles of these authors are articulated in the ‘author contributions’ section.”If your commercial affiliation did play a role in your study, please state and explain this role within your updated Funding Statement.  b. Please also provide an updated Competing Interests Statement declaring this commercial affiliation along with any other relevant declarations relating to employment, consultancy, patents, products in development, or marketed products, etc.   Within your Competing Interests Statement, please confirm that this commercial affiliation does not alter your adherence to all PLOS ONE policies on sharing data and materials by including the following statement: "This does not alter our adherence to  PLOS ONE policies on sharing data and materials.” (as detailed online in our guide for authors http://journals.plos.org/plosone/s/competing-interests) . If this adherence statement is not accurate and  there are restrictions on sharing of data and/or materials, please state these. Please note that we cannot proceed with consideration of your article until this information has been declared. Please include both an updated Funding Statement and Competing Interests Statement in your cover letter. We will change the online submission form on your behalf. 5. Thank you for stating the following in the Competing Interests section: YT, SS, and MS received financial support from MSD through their employment institutions for organizing and conducting the study, analyzing the collected data, and writing the report/manuscript. ML reports grants, honoraria, or consulting fee from MSD, Johnson&Johnson, Gebro Pharma, and Orpha Swiss. SU reports research grants, honoraria, or consulting fee from the Swiss National Science Foundation, Zurich and Swiss Lung League and EMDO foundation, Orpha Swiss, Janssen SA, MSD SA, Gebro SA, Ideogen and Astra Zeneca (all unrelated to the present work). SG reports grants, honoraria, or consulting fee from MSD, Johnson&Johnson, Gebro Pharma, and Orpha Swiss. AB reports grants, honoraria, or consulting fee from AstraZeneca, Sanofi, GSK, Novartis, OM pharma, Boehringer Ingelheim. MS reports grants, honoraria, or consulting fee from, AbbVie, Bristol-Myers Squibb, Novartis, Pfizer, Roche, and AstraZeneca. Other authors declare no conflicts of interest in relation to this manuscript.  Please confirm that this does not alter your adherence to all PLOS ONE policies on sharing data and materials, by including the following statement: "This does not alter our adherence to  PLOS ONE policies on sharing data and materials.” (as detailed online in our guide for authors http://journals.plos.org/plosone/s/competing-interests). If there are restrictions on sharing of data and/or materials, please state these. Please note that we cannot proceed with consideration of your article until this information has been declared.  Please include your updated Competing Interests statement in your cover letter; we will change the online submission form on your behalf. 6. We note that you have indicated that there are restrictions to data sharing for this study. For studies involving human research participant data or other sensitive data, we encourage authors to share de-identified or anonymized data. However, when data cannot be publicly shared for ethical reasons, we allow authors to make their data sets available upon request. For information on unacceptable data access restrictions, please see http://journals.plos.org/plosone/s/data-availability#loc-unacceptable-data-access-restrictions.  Before we proceed with your manuscript, please address the following prompts: a) If there are ethical or legal restrictions on sharing a de-identified data set, please explain them in detail (e.g., data contain potentially identifying or sensitive patient information, data are owned by a third-party organization, etc.) and who has imposed them (e.g., a Research Ethics Committee or Institutional Review Board, etc.). Please also provide contact information for a data access committee, ethics committee, or other institutional body to which data requests may be sent. b) If there are no restrictions, please upload the minimal anonymized data set necessary to replicate your study findings to a stable, public repository and provide us with the relevant URLs, DOIs, or accession numbers. Please see http://www.bmj.com/content/340/bmj.c181.long for guidelines on how to de-identify and prepare clinical data for publication. For a list of recommended repositories, please see https://journals.plos.org/plosone/s/recommended-repositories. You also have the option of uploading the data as Supporting Information files, but we would recommend depositing data directly to a data repository if possible. Please update your Data Availability statement in the submission form accordingly. 7. Please include captions for your Supporting Information files at the end of your manuscript, and update any in-text citations to match accordingly. Please see our Supporting Information guidelines for more information: http://journals.plos.org/plosone/s/supporting-information. 8. If the reviewer comments include a recommendation to cite specific previously published works, please review and evaluate these publications to determine whether they are relevant and should be cited. There is no requirement to cite these works unless the editor has indicated otherwise.

Reviewers' comments:

Reviewer's Responses to Questions

**Comments to the Author**

1. Is the manuscript technically sound, and do the data support the conclusions?

Reviewer #1: Yes

Reviewer #2: Yes

2. Has the statistical analysis been performed appropriately and rigorously? 

Reviewer #1: Yes

Reviewer #2: Yes

3. Have the authors made all data underlying the findings in their manuscript fully available?

The PLOS Data policy requires authors to make all data underlying the findings described in their manuscript fully available without restriction, with rare exception (please refer to the Data Availability Statement in the manuscript PDF file). The data should be provided as part of the manuscript or its supporting information, or deposited to a public repository. For example, in addition to summary statistics, the data points behind means, medians and variance measures should be available. If there are restrictions on publicly sharing data—e.g. participant privacy or use of data from a third party—those must be specified.requires authors to make all data underlying the findings described in their manuscript fully available without restriction, with rare exception (please refer to the Data Availability Statement in the manuscript PDF file). The data should be provided as part of the manuscript or its supporting information, or deposited to a public repository. For example, in addition to summary statistics, the data points behind means, medians and variance measures should be available. If there are restrictions on publicly sharing data—e.g. participant privacy or use of data from a third party—those must be specified.requires authors to make all data underlying the findings described in their manuscript fully available without restriction, with rare exception (please refer to the Data Availability Statement in the manuscript PDF file). The data should be provided as part of the manuscript or its supporting information, or deposited to a public repository. For example, in addition to summary statistics, the data points behind means, medians and variance measures should be available. If there are restrictions on publicly sharing data—e.g. participant privacy or use of data from a third party—those must be specified.requires authors to make all data underlying the findings described in their manuscript fully available without restriction, with rare exception (please refer to the Data Availability Statement in the manuscript PDF file). The data should be provided as part of the manuscript or its supporting information, or deposited to a public repository. For example, in addition to summary statistics, the data points behind means, medians and variance measures should be available. If there are restrictions on publicly sharing data—e.g. participant privacy or use of data from a third party—those must be specified.

Reviewer #1: Yes

Reviewer #2: Yes

4. Is the manuscript presented in an intelligible fashion and written in standard English?

Reviewer #1: Yes

Reviewer #2: Yes

5. Review Comments to the Author

**Reviewer #1:** The issue of medical costs in PH treatment is a major challenge that is a common topic around the world. I would like to point out a few issues.The issue of medical costs in PH treatment is a major challenge that is a common topic around the world. I would like to point out a few issues.The issue of medical costs in PH treatment is a major challenge that is a common topic around the world. I would like to point out a few issues.The issue of medical costs in PH treatment is a major challenge that is a common topic around the world. I would like to point out a few issues.

1.Annual cost estimates in this study are derived by extrapolating short observation periods, which introduces uncertainty that requires clearer justification and sensitivity analysis.

Patient-reported outcomes were collected for the preceding 4 weeks and annualized by multiplication, while healthcare utilization data from medical records covered 6 months and were annualized by doubling. Although commonly used in cost-of-illness studies, this approach assumes temporal stability of costs.

PAH is characterized by episodic exacerbations, hospitalizations, treatment changes, and possible seasonal effects; therefore, short-term observations may not be representative of average annual resource use. Extrapolation using fixed factors (e.g., ×13 or ×2) may lead to under- or overestimation.

To strengthen the robustness of the findings, I recommend clarifying the rationale for the chosen extrapolation factors and performing sensitivity analyses using alternative annualization assumptions (e.g., varying the 4-week factor within ×10–14). Demonstrating that the main conclusions remain consistent across these scenarios would substantially improve transparency and confidence in the annual cost estimates.

2.Cost estimates by disease severity, particularly for the ESC/ERS high-risk group, appear unstable due to very small subgroup sizes. In rare diseases such as PAH, mean costs can be strongly influenced by a few high-cost cases, increasing the risk of chance effects.

To improve robustness, I recommend reporting medians and measures of dispersion in addition to means, providing uncertainty estimates (e.g., bootstrap confidence intervals), and considering complementary analyses modeling disease severity as a continuous variable rather than relying solely on categorical stratification.

**Reviewer #2:** This reviewer has the following comments to be addressed.This reviewer has the following comments to be addressed.This reviewer has the following comments to be addressed.This reviewer has the following comments to be addressed.

1) In table 2, epoprostenol is lacked. In addition, p.o., i.v., s.c., or inhalation should be clearly stated. And, steroids and immunosuppressants in patients associated with connective tissue disease (CTD) are better to be added.

2) Did the presence or absence of i.v./s.c. prostacyclin analogues affect healthcare costs and national burden?

3) In CTD-associated PAH, SSc and non-SSc are clearly distiguished in ESC/ERS PH guideline. Did the difference between SSc vs. non-SSc affect healthcare costs and national burden in CTD-PAH patients? Furthermore, did the presence or absence of steroids and immunosuppressants affect in those patients?

4) Which of WHO-FC, ESC/ERS 2022 risk strata, PAH subgroup clinical classification, and the presence or absence of i.v./s.c. prostacyclin analogues had the greatest impact on healthcare costs/burden?

5) How are the results of Switzerland in this study expected to differ from those in the United States, Europe, and Asia?

6) Can the authors consider in more depth the clinical impact of this study results, its potential effect on reducing the burden on patients, and its impact on the future of PAH treatment?

6. PLOS authors have the option to publish the peer review history of their article (what does this mean?). If published, this will include your full peer review and any attached files.). If published, this will include your full peer review and any attached files.). If published, this will include your full peer review and any attached files.). If published, this will include your full peer review and any attached files.

...

Reviewer #1: No

Reviewer #2: No

---

## [Author Response · Author response to Decision Letter 1]

1 Apr 2026

Reviewer 1

The issue of medical costs in PH treatment is a major challenge that is a common topic around the world. I would like to point out a few issues.

1. Annual cost estimates in this study are derived by extrapolating short observation periods, which introduces uncertainty that requires clearer justification and sensitivity analysis.

Patient-reported outcomes were collected for the preceding 4 weeks and annualized by multiplication, while healthcare utilization data from medical records covered 6 months and were annualized by doubling. Although commonly used in cost-of-illness studies, this approach assumes temporal stability of costs.

PAH is characterized by episodic exacerbations, hospitalizations, treatment changes, and possible seasonal effects; therefore, short-term observations may not be representative of average annual resource use. Extrapolation using fixed factors (e.g., ×13 or ×2) may lead to under- or overestimation.

To strengthen the robustness of the findings, I recommend clarifying the rationale for the chosen extrapolation factors and performing sensitivity analyses using alternative annualization assumptions (e.g., varying the 4-week factor within ×10–14). Demonstrating that the main conclusions remain consistent across these scenarios would substantially improve transparency and confidence in the annual cost estimates.

Author’s response (AR): We appreciate the reviewer’s observation regarding the uncertainty introduced by annualizing short observation periods. We agree that extrapolating 4‑week patient‑reported outcomes and 6‑month healthcare utilization data assumes temporal stability and may not fully reflect the episodic nature of PAH. Annualization using factors of ×13 and ×2 is a standard approach in cost‑of‑illness studies and allows comparability with previous PAH economic evaluations [see for example https://pubmed.ncbi.nlm.nih.gov/27201788/, https://pubmed.ncbi.nlm.nih.gov/31493181/].As suggested by the reviewer, we added a sensitivity analysis in which the 4-week outcomes were annualized using factor x10-14. In parallel we also varied the 6-month outcomes by 20%. We believe these additions improve the transparency and robustness of the annual cost estimates and thank the reviewer for this helpful suggestion.

In the section Methods – Statistical analysis we added following text:” Considering the uncertainty associated with annualizing costs derived from patient-reported outcomes collected over relatively short observation periods (4 weeks or 6 months), sensitivity analyses were conducted. For variables collected over 4 weeks and extrapolated to one year, multiplication factors ranging from 10 to 14 were used, with 13 applied in the main analysis. For variables collected over a 6-month period and extrapolated to one year, the multiplication factor used in the main analysis was varied by ±20%.”

In the section Results – Estimated annual costs per patient we added following text:” In the sensitivity analysis, in which outcomes collected over 4 weeks and 6 months were annualized using lower and higher multiplication factors, estimated direct costs ranged from €87,114 to €130,227, indirect costs from €22,957 to €32,140, and total costs from €110,071 to €162,367. In general, the relative distribution of the costs components remained stable. Varying the multiplication factor for outcomes collected over a 4-week period had little impact on cost estimates. In contrast, varying the annualization of variables collected over a 6-month period (particularly treatment costs) had a moderate impact on the estimated costs (see S9 Table).”

In the section Discussion – Strength and limitations we added following text:” Fourth, some costs variables were based on patient-reported outcomes collected over a short period (4 weeks). Extrapolating these variables by simple multiplication implies the assumption that the impact of PAH on patient health remains constant over time. However, seasonal effects, which could lead, for example, to more frequent exacerbations, cannot be excluded. The sensitivity analysis conducted to assess the uncertainty related to the annualization of the costs suggested that the variation of the multiplication factor (10-14 instead of 13 used in the main analysis) had a minimal impact on the direct costs and a rather small impact on the indirect costs.”

In the Supporting information we added Table S7 illustrating the results of the sensitivity analysis.

2. Cost estimates by disease severity, particularly for the ESC/ERS high-risk group, appear unstable due to very small subgroup sizes. In rare diseases such as PAH, mean costs can be strongly influenced by a few high-cost cases, increasing the risk of chance effects. To improve robustness, I recommend reporting medians and measures of dispersion in addition to means, providing uncertainty estimates (e.g., bootstrap confidence intervals), and considering complementary analyses modeling disease severity as a continuous variable rather than relying solely on categorical stratification.

AR: We appreciate the reviewer’s comment regarding the instability of cost estimates in the ESC/ERS high‑risk subgroup. We agree that small sample sizes can make mean values sensitive to a few high‑cost cases.

We thank the reviewer for this important comment regarding the potential instability of cost estimates across disease severity strata, particularly in the ESC/ERS high-risk group with small sample sizes. In response, we have expanded the reporting of descriptive and uncertainty measures. Specifically, we added a new supplementary table (Table S5) presenting means, standard deviations, medians, interquartile ranges, selected percentiles, and bootstrapped 95% confidence intervals for all cost outcomes by disease severity group. These additional metrics allow a more comprehensive assessment of distributional characteristics and the influence of potential high-cost outliers, thereby improving the transparency and robustness of the reported estimates. We have also clarified this addition in the Results section and referenced the new table in the manuscript: “To assess the robustness of cost estimates across disease severity groups, additional distributional analyses were conducted. Supplementary Table X reports means, standard deviations, medians, interquartile ranges, selected percentiles, and bootstrapped 95% confidence intervals for all cost outcomes by ESC/ERS risk category. Cost distributions were right-skewed, with greater variability and wider confidence intervals observed in higher-risk groups, reflecting small subgroup sizes and the influence of high-cost cases. These results provide additional context for interpreting severity-stratified cost estimates.”

Concerning additional analyses using disease severity as continuous variable: we thank the reviewer for this thoughtful suggestion. Disease severity was analysed using WHO FC and ESC/ERS risk categories because these represent validated and guideline-based prognostic strata that directly inform treatment decisions and patient management in PAH. The WHO FC and ESC/ERS risk assessment is derived from multiple heterogeneous clinical variables and does not constitute a naturally continuous measure; therefore, modelling severity as a continuous variable would require assumptions regarding linearity and equal spacing between risk levels that are not established. In addition, given the limited sample size inherent to rare disease research and the skewed distribution of cost data, continuous modelling could introduce model instability and overfitting without necessarily improving interpretability. As the primary objective of this study was to compare economic burden across clinically meaningful severity groups, we considered categorical stratification to be the most appropriate and transparent analytical approach.

Reviewer 2

This reviewer has the following comments to be addressed.

1. In table 2, epoprostenol is lacked. In addition, p.o., i.v., s.c., or inhalation should be clearly stated. And, steroids and immunosuppressants in patients associated with connective tissue disease (CTD) are better to be added.

AR: We thank the reviewer for this valuable comment. Among the participating PAH patients, only one individual received epoprostenol in combination with macitentan, prescribed in the context of a late PAH diagnosis during pregnancy, where it represented the most appropriate therapeutic option. Epoprostenol is not included in the list of medications covered by mandatory basic health insurance in Switzerland, and no official Swiss public price is available. One option would have been to directly contact the pharmaceutical company to obtain the applicable cost. However, given that this concerns a single patient in our cohort, doing so could raise ethical concerns regarding patient rights. Therefore, for this patient, only macitentan costs were included in the analysis, while potential epoprostenol costs were omitted. Given that this concerns a single case, we expect this omission to have a limited impact on the overall cost estimates.

We added this information in the limitations.

We have now revised Table 2 to clearly indicate the routes of administration for all listed therapies.

Concerning steroids and immunosuppressant: in the present study, we specifically focused on medications indicated for the treatment of pulmonary arterial hypertension (PAH). Therapies prescribed for concomitant conditions, including steroids and immunosuppressants used in patients with connective tissue disease (CTD), were not within the scope of our analysis and were therefore not included. We have clarified this point in the limitations: “Seventh, this study focused exclusively on medications indicated for PAH and did not account for treatments related to concomitant diseases. Given that most participants were polymorbid, some healthcare resource use and productivity losses may have been attributable to comorbidities rather than PAH itself. Consequently, the reported costs may both underestimate the true economic burden—by excluding therapies required for associated conditions (e.g., steroids or immunosuppressants in patients with connective tissue disease)—and overestimate PAH-specific costs by capturing resource utilization driven by coexisting diseases”

2. Did the presence or absence of i.v./s.c. prostacyclin analogues affect healthcare costs and national burden?

AR: We thank the reviewer for this relevant comment. The impact of i.v./s.c. prostacyclin analogue therapy on healthcare costs and overall economic burden was already addressed in the manuscript through exploratory analyses. As reported in the Results section, treatment with intravenous treprostinil was identified as a major cost driver, with substantially higher direct and indirect annual costs observed among treated patients compared with those not receiving this therapy. These findings demonstrate that prostacyclin analogue use is strongly associated with increased healthcare expenditures and therefore meaningfully contributes to the overall budget impact.

3. In CTD-associated PAH, SSc and non-SSc are clearly distiguished in ESC/ERS PH guideline. Did the difference between SSc vs. non-SSc affect healthcare costs and national burden in CTD-PAH patients? Furthermore, did the presence or absence of steroids and immunosuppressants affect in those patients?

AR: We thank the reviewer for this valuable remark. In response, we identified that systemic sclerosis had inadvertently been omitted from the list of main comorbidities and have now added this information to Table 1, indicating that 19 patients with PAH had systemic sclerosis.

Among the 37 patients with CTD-associated PAH, 19 were classified as having systemic sclerosis (SSc). Compared with non-SSc CTD-PAH patients, those with SSc were more frequently female (84.2% vs. 61.1%) and older (mean age 69.8 vs. 62.4 years). Direct and indirect costs were higher in non-SSc patients (€139,080 and €44,153, respectively) compared with patients with SSc (€64,981 and €25,985, respectively). These findings suggest that differences in CTD subtype may influence healthcare costs; however, given the limited subgroup sizes, these observations should be interpreted cautiously.

We added following text in the Results section: “Another exploratory subgroup analysis was conducted among patients with connective tissue disease (CTD)–associated PAH to compare systemic sclerosis (SSc) and non-SSc cases. Of the 37 patients with CTD-associated PAH, 19 had SSc. Patients with SSc were more frequently female (84.2% vs. 61.1%) and older (mean age 69.8 vs. 62.4 years) compared with non-SSc patients. Mean annual direct and indirect costs were higher in non-SSc patients (€139,080 and €44,153, respectively) than in those with SSc (€64,981 and €25,985, respectively).”

As previously stated, the present study focused on PAH-related costs and did not collect detailed information on treatments prescribed for concomitant diseases. Consequently, the potential impact of steroid or immunosuppressant use in CTD-associated PAH patients could not be evaluated.

Which of WHO-FC, ESC/ERS 2022 risk strata, PAH subgroup clinical classification, and the presence or absence of i.v./s.c. prostacyclin analogues had the greatest impact on healthcare costs/burden?

AR: We thank the reviewer for raising this important question. To evaluate the relative impact of disease severity measures, clinical classification, and treatment characteristics on healthcare costs, we performed a generalized linear regression analysis including WHO functional class (WHO-FC), ESC/ERS 2022 risk strata, PAH subgroup clinical classification, use of intravenous treprostinil, age, and sex as covariates.

The factors with the greatest impact on total healthcare costs were treatment with treprostinil (coefficient 1.432, p<0.001) and advanced functional impairment, particularly WHO-FC IV (coefficient 0.970, p=0.035), followed by WHO-FC III (coefficient 0.745, p<0.001) and WHO-FC II (coefficient 0.379, p=0.022). In contrast, PAH clinical subtype and ESC/ERS 2022 risk strata were not significantly associated with total costs in the multivariable model.

These findings suggest that treatment intensity and functional status were stronger determinants of healthcare costs than etiological classification or categorical risk stratification in this cohort.

We would like to emphasize that these findings should be interpreted with caution. The regression analysis was conducted in a relatively small sample, which may limit statistical power and the stability of model estimates. In addition, several covariates included in the model, particularly WHO functional class and ESC/ERS risk strata, partially overlap conceptually and clinically, as both reflect disease severity. WHO functional class and ESC/ERS risk strata were in fact significantly correlated (correlation coefficient 0.575, p<0.001). This collinearity may have attenuated independent associations and should be considered when interpreting the relative contribution of individual predictors to healthcare costs.

In the Results section we added following text: “The second GLM aimed to investigate which factors among WHO-FC, ESC/ERS 2022 risk strata, and PAH clinical subgroup classification had the highest impact on total costs. Treatment with treprostinil was the strongest determinant of total costs (coefficient 1.432, p<0.001). Higher WHO-FC was also significantly associated with increased costs, with the largest effect observed for WHO-FC IV (coefficient 0.970, p=0.035), followed by WHO-FC III (coefficient 0.745, p<0.001) and WHO-FC II (coefficient 0.379, p=0.022). In contrast, PAH clinical subtype and ESC/ERS 2022 risk strata were not significantly associated with total healthcare costs in the multivariable model. It is important to emphasize that the regression analysis was conducted in a relatively small sample, which may limit statistical power and the stability of model estimates. In addition, several covariates included in the model, particularly WHO functional class and ESC/ERS risk strata, partially overlap conceptually and clinically, as both reflect disease severity. WHO-FC and ESC/ERS risk strata were in fact significantly correla

---

## [Decision Letter · Decision Letter 1]

13 Apr 2026

Economic Burden of Pulmonary Arterial Hypertension in Switzerland.

PONE-D-25-57450R1

Dear Dr. Tomonaga,

We’re pleased to inform you that your manuscript has been judged scientifically suitable for publication and will be formally accepted for publication once it meets all outstanding technical requirements.

Kind regards,

Yoshihiro Fukumoto

Academic Editor

PLOS One

Additional Editor Comments (optional):

Reviewers' comments:

Reviewer's Responses to Questions

**Comments to the Author**

1. If the authors have adequately addressed your comments raised in a previous round of review and you feel that this manuscript is now acceptable for publication, you may indicate that here to bypass the “Comments to the Author” section, enter your conflict of interest statement in the “Confidential to Editor” section, and submit your "Accept" recommendation.

Reviewer #1: All comments have been addressed

Reviewer #2: All comments have been addressed

2. Is the manuscript technically sound, and do the data support the conclusions?

Reviewer #1: Yes

Reviewer #2: Yes

3. Has the statistical analysis been performed appropriately and rigorously? 

Reviewer #1: Yes

Reviewer #2: Yes

4. Have the authors made all data underlying the findings in their manuscript fully available?

The PLOS Data policy requires authors to make all data underlying the findings described in their manuscript fully available without restriction, with rare exception (please refer to the Data Availability Statement in the manuscript PDF file). The data should be provided as part of the manuscript or its supporting information, or deposited to a public repository. For example, in addition to summary statistics, the data points behind means, medians and variance measures should be available. If there are restrictions on publicly sharing data—e.g. participant privacy or use of data from a third party—those must be specified.requires authors to make all data underlying the findings described in their manuscript fully available without restriction, with rare exception (please refer to the Data Availability Statement in the manuscript PDF file). The data should be provided as part of the manuscript or its supporting information, or deposited to a public repository. For example, in addition to summary statistics, the data points behind means, medians and variance measures should be available. If there are restrictions on publicly sharing data—e.g. participant privacy or use of data from a third party—those must be specified.requires authors to make all data underlying the findings described in their manuscript fully available without restriction, with rare exception (please refer to the Data Availability Statement in the manuscript PDF file). The data should be provided as part of the manuscript or its supporting information, or deposited to a public repository. For example, in addition to summary statistics, the data points behind means, medians and variance measures should be available. If there are restrictions on publicly sharing data—e.g. participant privacy or use of data from a third party—those must be specified.requires authors to make all data underlying the findings described in their manuscript fully available without restriction, with rare exception (please refer to the Data Availability Statement in the manuscript PDF file). The data should be provided as part of the manuscript or its supporting information, or deposited to a public repository. For example, in addition to summary statistics, the data points behind means, medians and variance measures should be available. If there are restrictions on publicly sharing data—e.g. participant privacy or use of data from a third party—those must be specified.

Reviewer #1: Yes

Reviewer #2: Yes

5. Is the manuscript presented in an intelligible fashion and written in standard English?

Reviewer #1: Yes

Reviewer #2: Yes

6. Review Comments to the Author

Reviewer #1: I believe that the revised manuscript has been carefully and thoroughly revised, and that all reviewer comments have been adequately addressed. I have no further comments.

Reviewer #2: (No Response)

7. PLOS authors have the option to publish the peer review history of their article (what does this mean?). If published, this will include your full peer review and any attached files.). If published, this will include your full peer review and any attached files.). If published, this will include your full peer review and any attached files.). If published, this will include your full peer review and any attached files.

...

Reviewer #1: No

Reviewer #2: No

---

## [Editor Report · Acceptance letter]

PONE-D-25-57450R1

PLOS One

Dear Dr. Tomonaga,

I'm pleased to inform you that your manuscript has been deemed suitable for publication in PLOS One. Congratulations! Your manuscript is now being handed over to our production team.

Kind regards,

on behalf of

Dr. Yoshihiro Fukumoto

Academic Editor

PLOS One